# Soil Aggregate Construction: Contribution from Functional Soil Amendment Fertilizer Derived from Dolomite

**Yaowei Zhan** [1,2], **Kaixin Jiang** [1,2], **Jiaquan Jiang** [1,2], **Lidan Zhang** [1,2], **Chengxiang Gao** [1,2], **Xiuxiu Qi** [1,2], **Jiayan Fan** [1,2], **Yuechen Li** [1,2], **Shaolong Sun** [1,2] and **Xiaolin Fan** [1,2,*]

1  College of Natural Resources and Environment, South China Agricultural University, Guangzhou 510642, China
2  R&D Center of Environment Friendly Fertilizer Science and Technology, South China Agricultural University, Guangzhou 510642, China
*  Correspondence: xlfan@scau.edu.cn

**Abstract:** Elastic and water stable macroaggregate are significant to soil structure. which is the base of the soil, to maintain sustainable agriculture. Whether and how functional amendment fertilizer is capable of construction of the macroaggregate is the main purpose of the study. Scanning electron microscopy (SEM) and energy dispersive spectroscopy (EDS) were used to investigate the effect of dolomite-based functional soil amendment fertilizers on soil structure. The fertilizers are beneficial to elastic-stable and water-stable aggregate construction. Calcined dolomite based soil amendment functional fertilizer (CDFF) was favorable to water-stable aggregates. The elastic-stable macroaggregate increased with lime, uncalcined dolomite based soil amendment functional fertilizer (UCDFF) and CDFF, and it was 3.0 to 4.2 times the microaggregate. The water-stable one of the CDFF was increased by 20.0%. The mean weight diameter (MWD) of the CDFF and the UCDFF increased by 0.05~0.19 mm, while that of lime only increased by 0.05 mm. The percentage of aggregate dispersion (PAD) of the CDFF was the least. SEM and EDS images revealed that Fe, Al, Si, Ca, Mg, C and O existed on the aggregates. The construction of stable aggregate lies in that the functional fertilizers can gradually neutralize soil $H^+$ and prevent soil colloid dispersion. Soil particles are bounded together to construct micro-agglomerates and then macro-agglomerates through $Ca^{2+}$, $Mg^{2+}$ bond bridge and $CaCO_3$, $MgCO_3$ salt bridge and adhesion of $SiO_2$, $Fe_2O_3$, $Al_2O_3$ as well as the other amorphous substances from the functional fertilizers.

**Keywords:** dolomite; functional soil amendment fertilizer; elastic- and water stable aggregate; percentage of aggregate dispersion; salt bridge





## 1. Introduction

Soil is a medium of crop growth and it is also a place where five soil fertility factors such as water, nutrients, aeration, heat and microorganisms are exchanged. High quality soil has good soil structure to ensure soil fertility [1], which is manifested as abundant soil elastic and water stable macro-aggregate. Good soil structures are not only characterized by coordination of the above five soil fertility factors, but also make the soil physical, chemical and biological processes suitable for crop growth [2,3] to maintain sustainable agriculture [4]. Food security depends on sustainable agriculture. The stability of soil structure is an important factor determining soil fertility and sustainable productivity, especially the stable aggregate because soil's physical, chemical and biological processes are influenced by soil structure [5]. Stability of the aggregates relates to the soil structure and physical and chemical properties [6]. Good soil structure is capable of coordinating the five fertility factors [7]. The quality of the soil structure is generally evaluated by the quantity of water-stable aggregates. The soil aggregate stability reflects their ability to resist various external forces without being destroyed. Good soil structure is not only characterized by mechanical stability, but also water stability to resist water erosion. The

quantity and distribution of water stable aggregate represent the stability of soil structure, water holding capacity, solution permeability and erosion resistance of the soil [8]. The water stable aggregate is very significant to stabilize soil structure and reduce water erosion in flooded or rainy seasons. Therefore, the construction of stable soil aggregates is of great significance for maintenance of the ideal soil porosity and water permeability under the action of rainfall-washing or tillage machinery force, and for the realization of sustainable agriculture.

Promoting soil aggregate construction through agricultural measures such as the application of soil amendments has become one of the key methods for sustainable agriculture and high yield production [9]. Studies have shown that soil aggregates can be destroyed by conventional tillage. Thus, non-tillage is normally used in agriculture to maintain soil structure. However, soil aggregates developed by such methods can be easily disintegrated with flooding [10]. The application of manure is a major technique to form water-stable soil aggregate and improve soil structure [11]. The advantages of organic fertilizer included the increase of soil nutrients, organic matter content and soil base ions concentration, which would be favorable to constructing water stable soil aggregates [12]. Calcium and magnesium ions in soil amendments also assisted the formation of water stable aggregates through the salt bridge [13].

Chemical fertilizer has greatly contributed to China's sustainable agriculture to meet its food consumption since the 1980s. However, the excessive and irrational application of chemical fertilizer [14], especially nitrogen fertilizer [15], has led to soil acidification in China for a long time [16]. Soil acidification not only causes the loss of calcium, magnesium and the other cation ions in the soil, but also leads to the destruction of the salt bridge among soil particles [17]. As a result, soil structure was deteriorated, soil was compacted [18] and the soil microbial diversity was poor [19,20]. Currently, common measures to neutralize soil acid are the application of lime [21], alkaline industrial and agricultural by-products [22] and soil amendments [23]. However, neutralizing soil acid with lime will induce soil compaction and soil structure deterioration [24]. The utilization of alkaline industrial and agricultural by-products will cause inevitable soil secondary pollution [25]. The efficacy of current soil amendments is also unpredictable because its quality is uneven or uncertain [26]. Therefore, we hold the opinion that the solution to the above problems is to develop and apply functional soil amendment fertilizer instead of conventional fertilizer. The function of the functional soil amendment fertilizer should supply nutrients on one hand and neutralize soil acid on the other hand. The fertilizer should also possess the ability to help construct good soil structure, and especially to promote the formation of the water stable aggregates. That is why dolomite-based Ca-Mg-Si functional soil amendment fertilizers have been developed on the base of our previous study about the dolomite based soil conditioner [27,28]. Therefore, the effects of the functional fertilizer on soil elastic- and water stable aggregate were tested through soil pot experiments with drying-wetting alternation and following under 25 °C constant temperature. The study aims to investigate whether and how the functional amendment fertilizer is capable of constructing the macroaggregate. The results will also expect to provide a theoretical basis to keep sustainable agriculture through improving soil, constructing stable aggregates and forming idea soil structures through the use of the functional fertilizer.

## 2. Materials and Methods

### 2.1. Soil and Fertilizer in Test

The soil was collected from Dongfanghong Farm, Leizhou Peninsula, Zhanjiang, Guangdong Province. The soil in test belongs to latosols developed from basalt parent materials, and it is orthic ferralsols according to the United Nations (UN) classification. The particle size of lime, natural dolomite and calcined dolomite was 0.15–0.25 mm. The dolomite in the test was collected from Jinfeng mine in Linwu, Chenzhou, Hunan Province, containing 33.12% CaO and 18.97% MgO. No other heavy metal was detected. The optimal grain size of the material was screened according to Jiang [27]. Soil mineral nitrogen

(Nmin, sum of ammonium and nitrate), available phosphorus, available potassium, exchangeable calcium and magnesium content were 21.6 mg/kg, 32.8 mg/kg, 64.7 mg/kg, 0.7142 cmol/kg and 0.0596 cmol/kg, respectively. The total amount of exchangeable acid was 3.4 cmol/kg and the pH value was 4.56.

There were two types of dolomite based calcium-magnesium-silicon functional compound fertilizer (namely functional amendment fertilizer), which was made of a mixture of $KNO_3$, tripling superphosphate and $K_2SO_4$ powder (0.5 mm) with calcined dolomite and uncalcined dolomite powder (<100 mesh) [27], respectively. The mass ratio of $N:P_2O_5:K_2O$ of the fertilizer was 1:0.5:1 and the mass ratio of dolomite to fertilizer is 6:4. Uncalcined dolomite-based functional amendment fertilizer was referred to the UCDFF, and the calcined dolomite-based functional amendment fertilizer was referred to the CDFF.

## 2.2. Experimental Design

An ANOVA two-way factorial design was used to investigate the effect of types and amount of the functional fertilizer on the soil structure by two laboratory soil pot experiments (namely Exp. 1 and Exp. 2) under the condition of drying-wetting alternation and fallow, respectively. Fertilizers of each experiment included the UCDFF, the CDFF and lime. Each fertilizer included three dosages, which were 4 g, 6 g and 8 g per kilogram of soil. The amount referred to the quantity of the lime or dolomite used in the functional fertilizers. The nitrogen, phosphorus and potassium content in each treatment were adjusted to the same level by $KNO_3$, $Ca(H_2PO_4)_2$ and $K_2SO_4$. The control treatment (CK) in the experiment was the one without the functional fertilizer or lime. There were 10 treatments for each experiment, and each treatment was tested in three replicates.

## 2.3. Culture Method

After fully mixing air-dried soil (through a 2 mm sieve) with the functional fertilizers and lime according to experiment design, 150 g of the mixture was put in a plastic cup (one replicate) of 5 cm in diameter and 9 cm height according to 1.45 of volume weight. The soil was watered to 65% of its field capacity with deionized water, and the cup was sealed with plastic film with 20 needle holes. All of the cups were then incubated in a constant temperature of 25 °C $\pm$ 1 °C, re-watering to 65% of the field capacity by weighing every three days. In Exp. 1, the soil in all of the treatments was left fallow for 30 days, and the soil was air-dried for 30 days after the cups were incubated for 15 days. The soil was then dried and wetted four times by adding deionized water to keep the soil water content at 65% of the field water holding capacity, and then dried naturally for one week. In Exp. 2, the soil received the same fallow treatment as above after 35 days of the incubation. The soil of each cup was then air-dried and tested.

## 2.4. Determination of Soil Aggregate Composition

The air-dried soil was broken apart by hand along the natural cracks from a bigger lump to a smaller one about 1 cm in diameter. The size distribution and composition of the soil aggregate were determined by the dry-sieving method [29].

Particle fraction of the elastic-stable aggregate was tested by the adoption of the Savinov dry sieve method as follows. One hundred grams of the soil lump with 1 cm diameter were measured and placed on the top of a set of sieve with a minimum diameter of 0.25 mm. The soil samples were shocked at a frequency of 30 times $min^{-1}$ for 5 min. The soil samples were then divided into two fractions and one part was soil >0.25 mm and the other was soil <0.25 mm. Each fraction of particle was calculated. Particles bigger than 0.25 mm were elastic-stable macroaggregate, and the one smaller than 0.25 mm was the elastic-stable microaggregate.

The water-stable aggregate determination adopted the Savinov wet sieve method. Fifty grams of the soil samples were prepared according to the particle fraction tested above and placed on the top of a sieve with a minimum diameter of 0.25 mm. The soil was immersed in 50 L deionized water for 5 min, and then a sequence of movements (a

frequency of 30 times min$^{-1}$) was carried out with the same set of sieves for 5 min. The soil left on the top of the sieve was dried at 50 °C until a constant weight was reached. Particles bigger than 0.25 mm were water-stable macroaggregate, and the ones smaller than 0.25 mm were water instable microaggregate. The aggregate stability was evaluated by mean weight diameter (MWD), geometric mean diameter (GMD) [30], and percentage of aggregate dispersion (PAD) [31]. They were calculated as follows.

$$MWD = \sum_{i=1}^{n}(x_i w_i) / \sum_{i=1}^{n} w_i \tag{1}$$

$$GWD = \exp\left(\sum_{i=1}^{n}(w_i \log x_i) / \sum_{i=1}^{n} w_i\right) \tag{2}$$

$$PAD = [(DR_{>0.25} - WR_{>0.25})/DR > 0.25]\, 100\% \tag{3}$$

where, Xi was the average diameter of aggregate in class i, Wi was the mass percentage of the aggregates in class i. $DR_{>0.25}$ was the ratio of elastic-stable aggregate with a diameter greater than 0.25 mm. $WR_{>0.25}$ was the ratio of water-stable aggregate with a diameter greater than 0.25 mm.

The ultrastructure of the soil aggregates was observed by use of a scanning electron microscope (SEM) and an energy dispersive spectrometer (EDS) (ZEISS EVO 10). Sand and gravel in the soil were removed before preparing the observation according to Wei [32].

## 3. Results and Analysis

### 3.1. Composition of Elastic-Stable Macro- and Microaggregate

The elastic-stable macroaggregate of Exp. 1 and Exp. 2 was 1.6 times and 2.0 times (1.8 on average) higher than that of microaggregates (Table 1), respectively. The elastic-stable macro-aggregates of lime, UCDFF and CDFF treatment of the two Exp. 1 and 2 were 2.6–3.0 and 2.4–3.2 (2.5–2.9 in average), 3.2–4.1 and 3.0–3.6 (3.2–3.9 in average) and 3.3–4.8 and 2.8–3.8 times (3.0–4.2 in average) that of microaggregates, respectively. Therefore, the soil was mainly composed of elastic-macroaggregate, which implied that the soil structure was elastic stable. It was also shown that the elastic stable macroaggregate was significantly more than that of the microaggregates in the soil. The number of the elastic stable macroaggregate of the CDFF, the UCDFF and lime treatment, especially the CDFF and the UCDFF treatment, was significantly greater than that of the control. The microaggregates of the control were the maximum among all treatments. The results indicated that the application of the functional amendment fertilizers was beneficial to constructing the elastic stable macroaggregate.

### 3.2. Composition of Water Stable Aggregate and Water Instable Aggregate

The number of water-stable macroaggregates (>0.25 mm) was significantly less than that of water instable microaggregates (<0.25 mm) in two experiments (Table 2). The microaggregates were 1.8~1.9 times (1.8 in average) higher than that of macroaggregates in the CK. The corresponding data of the lime, UCDFF and CDFF were 1.0~1.6 (1.3 in average), 1.0~1.6 (1.3 in average) and 0.9~1.4 times (1.1 in average), respectively. Results indicated that the water-stable macroaggregate would be destroyed to some extent under the condition of flooding, which was not conducive to constructing a good or stable soil structure. However, the microaggregates proportion in the CDFF treatment was the least among the four treatments, which implied that the CDFF was beneficial to constructing water-stable aggregates. However, the water-stable macroaggregates were significantly greater than that of the unstable microaggregates when the amount of CDFF reached 8 g/kg of soil. Thus, the CDFF was favorable to construct water stable soil structures. In other words, when the dosage of the functional amendment fertilizer reached 8 g/kg, the fertilizer was able to help water stable aggregates and soil structure formation.

**Table 1.** Effects of the functional amendment fertilizers and lime on the composition of soil elastic stable macroaggregate (>0.25 mm) and microaggregate (<0.25 mm).

| Treatment | Composition of Macro- and Microaggregate (%) | | | |
| --- | --- | --- | --- | --- |
| | Exp. 1 | | Exp. 2 | |
| | >0.25 mm | <0.25 mm | >0.25 mm | <0.25 mm |
| CK | 61.2 ± 0.4 d | 38.8 ± 0.4 a ** | 61.2 ± 1.4 e | 38.8 ± 1.4 a ** |
| Lime 4 | 72.3 ± 0.6 c | 27.7 ± 0.6 bc ** | 72.3 ± 2.7 abc | 27.7 ± 2.7 d ** |
| Lime 6 | 72.4 ± 1.1 c | 27.6 ± 1.1 bc ** | 72.4 ± 2.6 d | 27.6 ± 2.6 bc ** |
| Lime 8 | 74.8 ± 1.8 bd | 25.2 ± 1.8 bc ** | 74.8 ± 1.2 cd | 25.2 ± 1.2 c ** |
| UCDFF 4 | 77.6 ± 1.5 ab | 22.4 ± 1.5 cd ** | 77.6 ± 1.3 c | 22.4 ± 1.3 cd ** |
| UCDFF 6 | 76.2 ± 1.3 bc | 23.8 ± 1.3 cd ** | 76.2 ± 0.0 ab | 23.8 ± 0.0 d ** |
| UCDFF 8 | 80.5 ± 1.1 ab | 19.5 ± 1.1 de ** | 80.5 ± 0.9 ab | 19.5 ± 0.9 de ** |
| CDFF 4 | 76.6 ± 0.1 bc | 23.4 ± 0.1 c ** | 76.6 ± 1.9 c | 23.4 ± 1.9 bc ** |
| CDFF 6 | 79.7 ± 1.8 ab | 20.3 ± 1.8 c ** | 79.7 ± 0.2 a | 20.3 ± 0.2 de ** |
| CDFF 8 | 82.9 ± 0.4 ab | 17.1 ± 0.4 e ** | 82.9 ± 2.1 a | 17.1 ± 2.1 e ** |

Note: Different lowercase letters in the same column indicated a significant difference among treatments by Duncan's multiple comparison ($p \leq 0.05$). ** indicated that there was significant difference between the macroaggregates (>0.25 mm) and microaggregate (<0.25 mm) of each treatment and experiment, respectively ($p \leq 0.01$, tested by Student *T*-test). CK = control; Lime = lime treatment; UCDFF = uncalcined dolomite based soil amendment functional fertilizer; CDFF = Calcined dolomite based soil amendment functional fertilizer; 4, 6, 8 = 4 g, 6 g and 8 g per kilogram of soil.

**Table 2.** Influence of the functional amendment fertilizers and lime on the composition of soil water-stable (>0.25 mm) and instable aggregate (<0.25 mm).

| Treatment | Composition of Macro- and Microaggregate (%) | | | |
| --- | --- | --- | --- | --- |
| | Exp. 1 | | Exp. 2 | |
| | >0.25 mm | <0.25 mm | >0.25 mm | <0.25 mm |
| CK | 35.7 ± 0.2 e | 64.3 ± 0.2 a ** | 34.6 ± 0.6 e | 65.4 ± 0.6 a ** |
| Lime 4 | 34.6 ± 0.6 e | 65.4 ± 0.6 a ** | 39.2 ± 2.3 abc | 60.8 ± 2.3 ab * |
| Lime 6 | 45.8 ± 3.7 c | 54.2 ± 3.7 c | 42.5 ± 2.7 d | 57.5 ± 2.7 bc * |
| Lime 8 | 49.5 ± 0.9 a | 50.5 ± 0.9 d | 48.9 ± 1.3 cd | 51.1 ± 1.3 cd |
| UCDFF 4 | 38.9 ± 3.4 d | 61.1 ± 3.4 ab ** | 40.3 ± 3.0 c | 59.7 ± 3.0 ab ** |
| UCDFF 6 | 39.5 ± 1.3 d | 60.5 ± 1.3 ab ** | 42.8 ± 3.3 ab | 57.2 ± 3.3 bc ** |
| UCDFF 8 | 50.1 ± 2.5 a | 49.9 ± 2.5 d | 47.7 ± 4.4 ab | 52.3 ± 4.4 cd |
| CDFF 4 | 42.6 ± 1.4 c | 57.4 ± 1.4 bc * | 42.1 ± 2.1 c | 57.9 ± 2.1 bcd ** |
| CDFF 6 | 49.2 ± 2.1 b | 50.8 ± 2.1 d | 49.0 ± 2.2 a | 51.0 ± 2.2 cd |
| CDFF 8 | 51.9 ± 0.7 a | 48.1 ± 0.7 d | 53.5 ± 1.8 a | 46.5 ± 1.8 d * |

Note: Different lowercase letters in the same column indicated a significant difference among treatments by Duncan's multiple comparison ($p \leq 0.05$). ** and * indicated the difference between the macroaggregates (>0.25 mm) and microaggregate (<0.25 mm) of each treatment and experiment at level of $p \leq 0.01$ and $p \leq 0.05$ (by Student's *t*-test), respectively. CK = control; Lime = lime treatment; UCDFF = uncalcined dolomite based soil amendment functional fertilizer; CDFF = Calcined dolomite based soil amendment functional fertilizer; 4, 6, 8 = 4 g, 6 g and 8 g per kilogram of soil.

### 3.3. Effects of Type and Dosage of Functional Fertilizer on Elastic-Stable Macro- and Microaggregate

3.3.1. Influence on the Elastic Stable Macroaggregate

The functional fertilizers and lime could significantly increase the elastic macroaggregate in the soil. The higher amount of the fertilizers added, the more aggregates there are in the soil (Figure 1). The elastic macroaggregates of lime, UCDFF and CDFF in comparison with the CK were increased by 19.6% and 10.4%, 27.7% and 16.5% and 27.7% and 16.5% in Exp. 1 and Exp. 2, respectively. However, the elastic-stable microaggregate of lime, UCDFF and CDFF compared with the CK were reduced by 30.9% and 20.2%, 43.6% and 31.5% and 47.8% and 32.1% in Exp. 1 and Exp. 2, respectively (Figure 1).

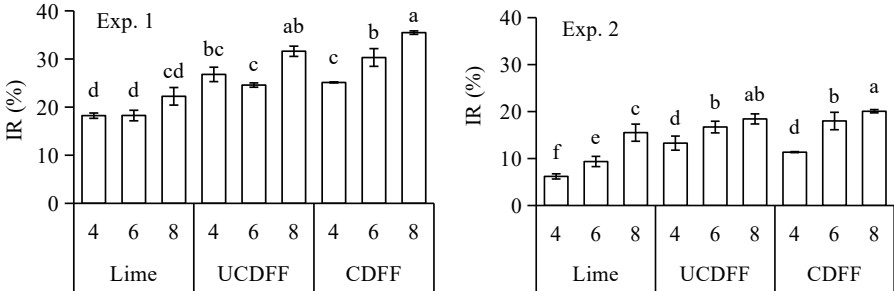

**Figure 1.** Effect of the functional amendment fertilizers and lime on the increase rate (IR) of soil elastic stable aggregates. Duncan's method was used for multiple comparisons. Lowercase letters indicate differences of IR among treatments. Lime = lime treatment; UCDFF = uncalcined dolomite based soil amendment functional fertilizer; CDFF = Calcined dolomite based soil amendment functional fertilizer; 4, 6, 8 = 4 g, 6 g and 8 g per kilogram of soil.

The amount of the macroaggregates was raised significantly with the dosage of lime and two functional fertilizers. In particular, when the lime was raised from 4 to 6 g/kg, the macroaggregate was increased by 1.6%, and from 4 to 8 g/kg, by 6.7%. The macroaggregate of UCDFF treatment was increased by 0.6 and 5.0 %when its dosage was raised from 4 to 6 g/kg and 4 to 8 g/kg, respectively. The corresponding data of the CDFF was 5.9 and 9.5%, individually (Figure 1). In other words, the elastic macroaggregates were significantly increased and the microaggregates decreased with the amount of lime, UCDFF and CDFF. When their dosage was increased to 8 g/kg, their efficacy was the greatest. Among them, the CDFF had the best effect on the increase of the elastic stable macroaggregate. Therefore, the application of the functional amendment fertilizers was beneficial to constructing a good elastic stable soil structure.

### 3.3.2. Influence on the Water Stable Macroaggregate

The application of the functional fertilizers and lime could significantly increase soil water-stable macroaggregates and decrease microaggregates (Figure 2) [33]. The water-stable macroaggregates of the lime, UCDFF and CDFF treatment were increased by 14.8% and 13.4%, 11.7% and 13.6% and 20.0% and 20.5% compared with the CK in Exp. 1 and Exp. 2 (Figure 2), respectively. The microaggregates of the lime, UCDFF and CDFF treatment were increased by 14.1% and 13.6%, 11.2% and 13.7%, 19.0% and 20.7% compared to those of CK in Exp. 1 and Exp. 2 (Figure 2), individually.

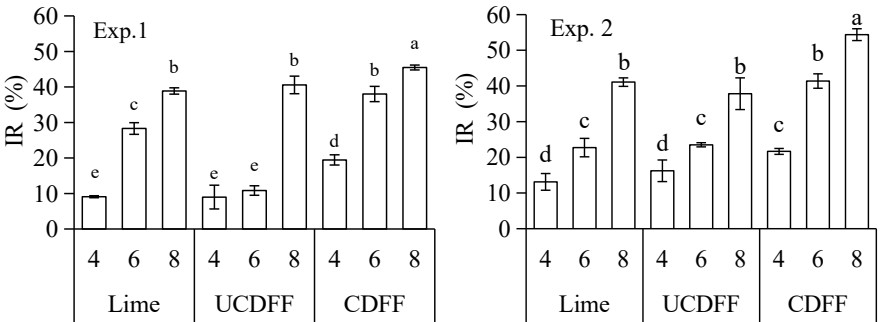

**Figure 2.** Effect of functional amendment fertilizers on increase rate (IR) of water-stable aggregate. Duncan's method was used for multiple comparisons. Lowercase letters indicate differences of IR among treatments. Lime = lime treatment; UCDFF = uncalcined dolomite based soil amendment functional fertilizer; CDFF = Calcined dolomite based soil amendment functional fertilizer; 4, 6, 8 = 4 g, 6 g and 8 g per kilogram of soil.

The microaggregates were decreased significantly with the dosage of lime and two functional fertilizers (Figure 2). When the lime dosage was increased from 4 to 6 g/kg

and 4 to 8 g/kg, the decrease rate (DR) of the microaggregates was increased by 3.1% and 12.3% in Exp. 1 and Exp. 2, individually. As the dosage of the UCDFF increased from 4 to 6 g/kg and 4 to 8 g/kg, the DR raised by 1.6% and 8.8% in Exp. 1 and Exp. 2, respectively. With the CDFF dosage enlarged from 4 to 6 g/kg and 4 to 8 g/kg, the DR values increased by 10.5% and 16.6% in Exp. 1 and Exp. 2, separately. Clearly, utilization of the lime and functional amendment fertilizers was able to increase the water-stable macroaggregates and decrease microaggregates significantly. When their dosage was increased to 8 g/kg, their effect on water-stable aggregate increment reached the maximum. Among the lime, UCDFF and CDFF, the CDFF works well. Therefore, fertilization of the functional soil amendment fertilizers is an effective measure to construct good soil structure with water stability.

*3.4. Effects of the Functional Amendment Fertilizer on MWD, GWD and PAD of Soil Aggregate*

3.4.1. Effects on the MWD and GMD of Elastic-Stable Aggregate

Mean weight diameter (MWD) of soil aggregate is an overall indicator of soil aggregate structure. The bigger the MWD value is, the better the soil aggregate structure is [34]. The geometric mean diameter (GWM) is an important indicator of soil aggregate stability, and the larger the GWM is, the more stable the soil aggregate is [35,36]. The efficacy of the lime and two functional fertilizers on the MWD and GMD was in the order of CDFF > UCDFF > Lime > CK (Table 3). The MWD of the CDFF and the UCDFF treatments was increased by 0.05–0.19 mm, while that of the lime was increased only by 0.05 mm. Thus, fertilization functional soil amendment fertilizer, specifically CDFF, can significantly improve and increase the MWD and the GWM of the elastic stable macroaggregate.

**Table 3.** Effect of the functional amendment fertilizers and lime on the MWD and GWD of the elastic stable macroaggregate.

| Experiment | Treatment | MWD (mm) | | GWD (mm) | |
|---|---|---|---|---|---|
| Exp. 1 | CK | $0.71 \pm 0.010$ | b | $0.57 \pm 0.002$ | c |
| | Lime | $0.64 \pm 0.019$ | c | $0.56 \pm 0.014$ | cd |
| | UCDFF | $0.72 \pm 0.029$ | ab | $0.61 \pm 0.022$ | ab |
| | CDFF | $0.75 \pm 0.010$ | a | $0.62 \pm 0.011$ | a |
| Exp. 2 | CK | $0.71 \pm 0.004$ | b | $0.59 \pm 0.003$ | a |
| | Lime | $0.69 \pm 0.010$ | c | $0.59 \pm 0.009$ | a |
| | UCDFF | $0.71 \pm 0.006$ | b | $0.59 \pm 0.004$ | a |
| | CDFF | $0.72 \pm 0.001$ | a | $0.59 \pm 0.004$ | a |

Note: Different lowercase letters in the same column indicated significant difference among treatments of each experiment by Duncan multiple range tests ($p \leq 0.05$). CK = control; Lime = lime treatment; UCDFF = uncalcined dolomite based soil amendment functional fertilizer; CDFF = Calcined dolomite based soil amendment functional fertilizer.

3.4.2. Effects on the MWD and GMD of Water Stable Aggregate

The MWD of water-stable macroaggregates of the CDFF and the UDFF treatments was significantly greater than that of lime and the CK (Table 4). However, the MWD of lime treatment was significantly lower than that of CK, which indicated that the lime was not conducive to raise the water stable aggregate. Therefore, the dolomite based functional amendment fertilizers, especially the CDFF, could significantly increase the MWD of water-stable macroaggregates. The GMD of the CDFF and the UDFF treatments in Exp. 1 was also remarkably higher than that of lime and the CK. There was no significant difference in GMD treated with CDFF and UDFF. In Exp. 2, there was no statistical difference of the GWD among the four treatments. In conclusion, the application of the functional fertilizers, especially the CDFF, can significantly increase the MWD and the GMD of the water stable aggregate and then enhance the water stability of the soil structure.

**Table 4.** Effect of the functional amendment fertilizers and lime on the MWD and GWD of the water stable aggregate.

| Experiment | Treatment | MWD (mm) | GWD (mm) |
|---|---|---|---|
| Exp. 1 | CK | 0.82 ± 0.019 d | 0.70 ± 0.003 d |
| | Lime | 0.90 ± 0.010 c | 0.76 ± 0.015 c |
| | UCDFF | 0.92 ± 0.023 b | 0.78 ± 0.015 b |
| | CDFF | 0.95 ± 0.005 a | 0.82 ± 0.013 a |
| Exp. 2 | CK | 0.80 ± 0.005 d | 0.69 ± 0.001 d |
| | Lime | 0.82 ± 0.036 c | 0.71 ± 0.028 c |
| | UCDFF | 0.99 ± 0.005 b | 0.82 ± 0.002 b |
| | CDFF | 1.00 ± 0.016 a | 0.84 ± 0.016 a |

Note: Different lowercase letters in the same column indicated a significant difference among treatments of each experiment by Duncan's multiple range tests ($p \leq 0.05$). CK = control; Lime = lime treatment; UCDFF = uncalcined dolomite based soil amendment functional fertilizer; CDFF = Calcined dolomite based soil amendment functional fertilizer.

### 3.4.3. Effect on Dispersion of the Aggregate

The percentage of aggregate dispersion (PAD) reflects the stability of the aggregate in water and the larger the PAD value is, the worse the stability of the aggregate is, and vice versa. There was no significant difference of the PAD among the UCDFF, lime (under 4 and 6 g/kg), CDFF (under 4 g/kg) and the CK treatments (Figure 3). Under 6 g/kg, the PAD of the CDFF treatment was remarkably less than that of the UCDFF and lime. The minimum PAD could be observed in the lime and the UCDFF treatments when the dosage was as high as 8 g/kg. However, the least PAD was found at 6 g/kg in the CDFF treatment. The PAD would not be significantly reduced with the increase of the CDFF to 8 g/kg. In other words, to realize the least dispersion of the aggregate, the minimum dosage of the lime and the UCDFF was 8 g/kg soil and the CDFF was only 6 g/kg. The results further confirmed that the CDFF was capable of increasing the soil water-stable aggregate.

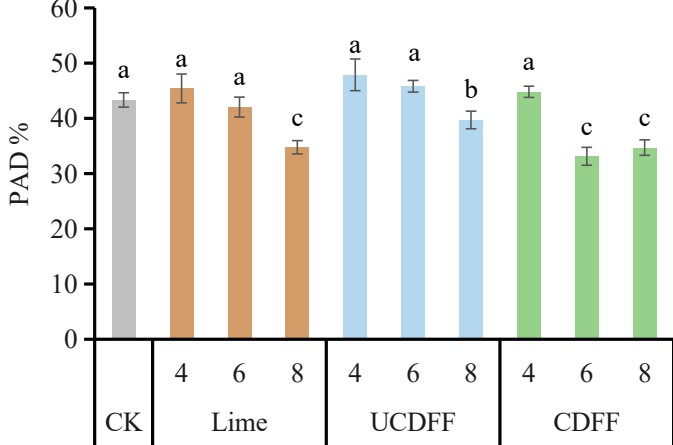

**Figure 3.** Effect of the functional fertilizers and lime on percentage of aggregate dispersion. PAD = percentage of aggregate dispersion; CK = control; Lime = lime treatment; UCDFF = uncalcined dolomite based soil amendment functional fertilizer; CDFF = Calcined dolomite based soil amendment functional fertilizer; 4, 6, 8 = 4 g, 6 g and 8 g per kilogram of soil. Duncan's method was used for multiple comparisons. Lowercase letters indicate differences of PAD among treatments.

### 3.5. Effect of the Functional Fertilizers and Lime on Soil Aggregate Microstructure

SEM images and EDS results of the elastic-stable micro- and macroaggregate of the CK, lime, UCDFF and CDFF treatments were shown in Figure 4. The outline, porous structure of the aggregates and binding state of soil particles could be observed clearly from the images. In the CK treatment, both elastic-stable microaggregates (Figure 4a) and macroaggregate (Figure 4b) were complete block structure. There was no obvious

accumulation of soil particles or aggregates. Only small, partially bonded clods could be observed. The clods were compact without porous structure. In lime treatment, both micro- and macroaggregates were formed through the accumulation of soil particles. However, the soil particles appeared to be bonded very tightly by melting substances. Most of the pores between the particles were filled by the melting substances. This phenomenon was more obvious in microaggregates (Figure 4a). This melting substance might be amorphous $CaCO_3$, $MgCO_3$, $Fe_2O_3$, $Al_2O_3$ and $SiO$, which was formed in the soil after the application of the lime. The accumulation of soil particles in micro- and macroaggregates of the UCDFF treatment was more obvious than that of the lime one. Soil particles in the UCDFF aggregates were not as tightly packed as that in the lime. There were obvious pores and fissures in the aggregates. The SEM images clearly illustrated that the micro- and macroaggregates of the soil treated by the CDFF were constructed by the accumulation of soil particles (Figure 4a,b). In different fertilizer treatments, the aggregates of the CDFF treatment were the only ones in which soil particles were distinctly embedded structure. In the embedded structure the soil particles were in close contact but were not dense. The pores between particles in the microaggregates were small, and the pores between the particles in the macroaggregates were large. It could be concluded that the application of the CDFF would promote the agglomerates construction through the embedded structure by large and small particles. Micro-pores existed in the micro-agglomerates and marco-pores were observed between macro-agglomerates. EDS test indicated that the main elements of the aggregates were Fe, Al, Si, Ca, Mg metal ions and C, O nonmetal elements. The element analysis proved that the mechanism of aggregates construction by the functional amendment fertilizers lies in $Ca^{2+}$, $Mg^{2+}$ bond bridge and $CaCO_3$, $MgCO_3$ salt bridge and adhesion of $SiO$ or $SiO_2$, $Fe_2O_3$, $Al_2O_3$ as well as the other amorphous substances.

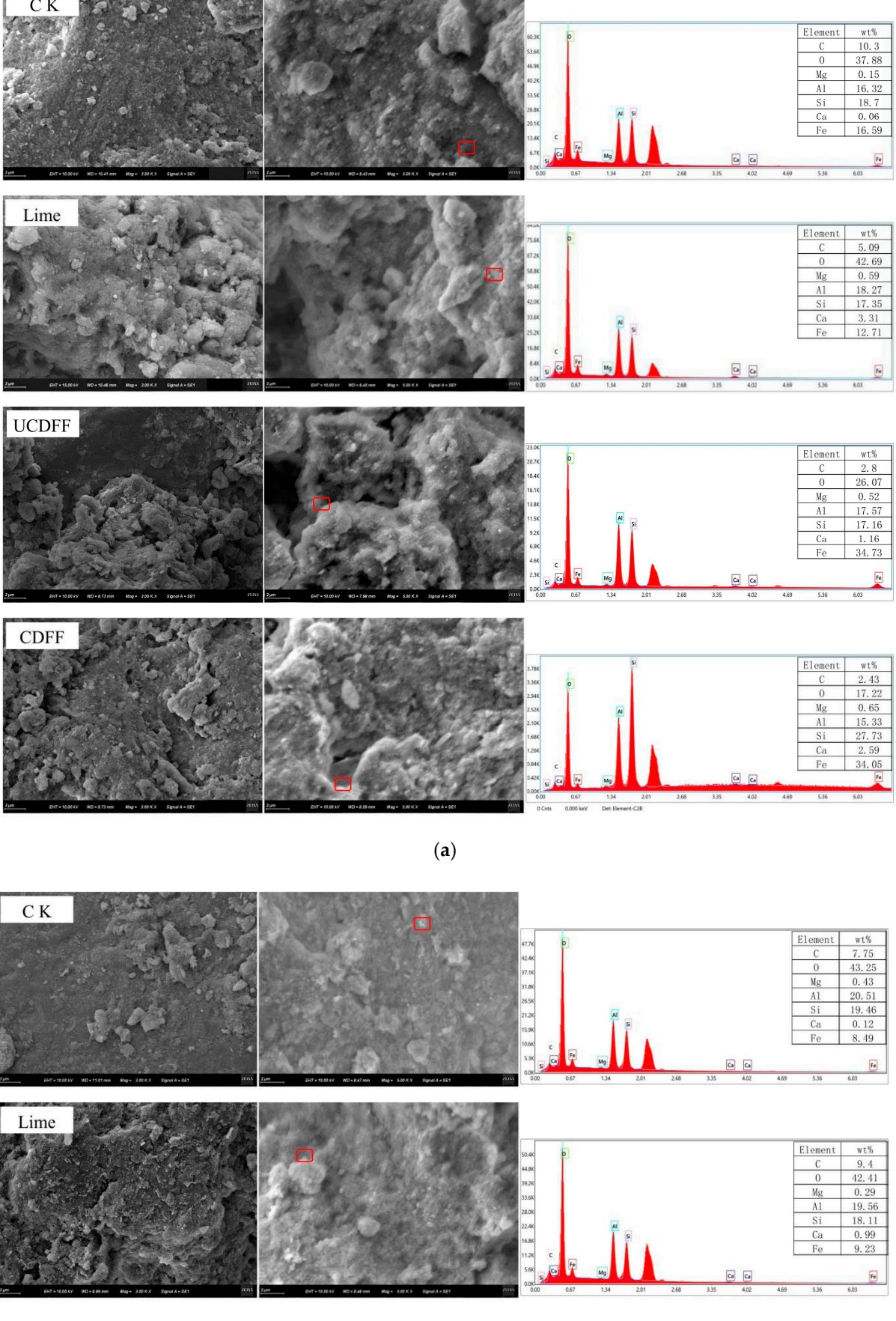

**Figure 4.** *Cont.*

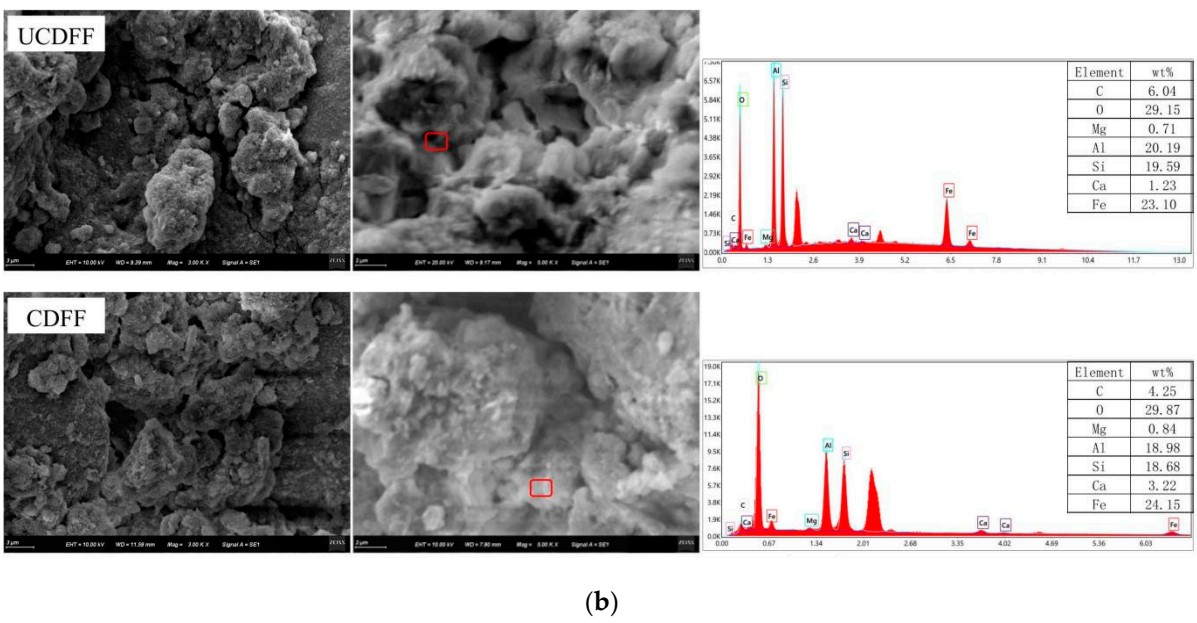

**(b)**

**Figure 4.** SEM images and EDS test results of soil aggregates (wt% = weight percent). (**a**) Microaggregates of air dried soil; (**b**) Macroaggregates of air dried soil.

## 4. Discussion

Studies have shown that agglomerate construction depends on organic matter, metal ions and their oxides in soil [37]. The function of organic matter is to bond soil particles together to build up macroaggregates by their functional groups. Metal ions have obvious positive or negative effects on soil aggregates. In soil, the radius of hydrated monovalent cations such as $K^+$, $Na^+$ is big and their existence will prevent soil particle flocculation [38,39]. Those ions will not only reduce the cohesiveness of soil colloids, but disperse the aggregate structure as well. The great amount of $K^+$ and $Na^+$ in the soil indicates the high salt concentration of the soil. The soil organic matter accumulation will be influenced under the high salt conditions because the micro-organism community and activity are influenced, which is not conducive to construct stable soil structure [40]. Cations in the soil also affect the formation and dispersion of soil aggregates. In general, $Na^+$ saturated soil will cause the expansion of soil particles and cause soil aggregates to disperse due to the hydration of $Na^+$. However, $K^+$ and other divalent cations such as $Ca^{2+}$ and $Mg^{2+}$ were beneficial to the structural improvement of soils. For soils dominated by kaolinite clay minerals, the cation adsorption capacity was as follows: $Ca^{2+} \geq Mg^{2+} > K^+$. For soils dominated by illite, the adsorption capacity in in the order of $K^+ > Ca^{2+} > Mg^{2+}$ [41]. The soil in this test is a latosol developed from basalt, which has a strong binding force to $Ca^{2+}$ and $Mg^{2+}$. After application of the alkaline functional fertilizers, Ca and Mg of the fertilizers will be better for the structural improvement of soils. In theory, the application of this fertilizer will also be better for the structural improvement of saline soil because the alkaline functional fertilizer contains a large amount of Ca and Mg to exchange the $Na^+$ in the soil particles. Therefore, as results of application the functional fertilizer dispersion of soil aggregates and the swelling of clay are prevented [42]. However, polyvalent cations such as $Ca^{2+}$ and $Mg^{2+}$ can not only promote the formation of soil aggregates through their own flocculation, but also maintain the stability of soil structure through ionic bond bridging [43,44] or the $CaCO_3$ and $MgCO_3$ salt bridge [45]. These two cations can also form organic-inorganic complexus in soil through clay-polyvalent cation-organic matter (C-P-OM) interactions. More stable soil aggregates are then formed through the accumulation of the complex and particles [30]. The results of this study (Figure 4) are consistent with the above research results. The efficacy of the CDFF treatment was better than that of the UCDFF due to the difference of dolomite added in them. Jiang [27] reported

that calcination could significantly increase the total alkalinity and short-term cumulative alkalinity of the dolomite. After the dolomite is calcined, the carbonate in the dolomite is transformed into $CO_2$ and released into the air. The $CaCO_3$ and $MgCO_3$ in the ore are transformed into $CaO$ and $MgO$. Relatively, the molar number of alkaline substances of the same mass of dolomite increases after calcination. The $MgO$ and $CaO$ of calcined dolomite will react with soil water to produce $Mg(OH)_2$ and $Ca(OH)_2$, which contain a large amount of $OH^-$ and high short-term cumulative alkalinity. They are obviously capable of treating or improving soil acid. In addition, $Mg(OH)_2$ and $Ca(OH)_2$ can gradually release calcium and magnesium while neutralizing soil acidity. It provides calcium and magnesium nutrients for soil. Therefore, the CDFF is much better than UCDFF.

Metal oxides, especially Fe/Al oxides, are main factors affecting the construction of the aggregates and its stability because Fe and Al oxides can form amorphous $Fe(OH)_3$ and $Al(OH)_3$ colloids in the soil. $Fe(OH)_3$, $Al(OH)_3$ and SOM will promote inorganic-clay particle-organic complex formations in the soil [46]. Six et al. reported that iron and aluminum oxides adsorbed on the surface of organic matter will bond to clay minerals by electrostatic action and act as bridges between soil particles [47]. The results in Table 1 showed that the macroaggregates could be significantly increased through the utilization of the functional amendment fertilizers (rich in calcium, magnesium and silicon) or lime (rich in calcium) of the soil, but the microaggregates declined. Those results are consistent with the literature reports above. Therefore, the soil aggregate was promoted through the bond bridge of the $Ca^{2+}$ from the functional fertilizers. After utilization of the functional fertilizers in acid soil, $Ca^{2+}$ of the fertilizer will compete with $Na^+$ and $K^+$ in soil for adsorption sites, and thus prevent $Na^+$ and $K^+$ from dispersing soil particles or aggregates.

It has been reported that Fe/Al changed from free oxides to chelated or amorphous Fe/Al oxides with the increase of soil pH value. Deprotonation of the amorphous oxides and the generation of functional groups on the interface were promoted, which then contributed to the construction of the soil aggregate structure [48]. The results of this paper are consistent with those reports, and the results have also directly been verified by the SEM observations of the soil micro-structure (Figure 4). From the SEM images, it was observed that the application of the functional fertilizers and lime could make the soil particles stick and agglomerate into larger soil particles, and gradually form large aggregates. The greater the amount of the functional fertilizers is, the more obvious the agglomeration efficacy is. The agglomeration effect of the functional fertilizers was more obvious than that of the lime (Figures 1 and 2) because they do not only contain abundant calcium but also magnesium and silicon. $Ca(OH)_2$ would be produced when the lime reacts with soil water. $H^+$ in soil was neutralized gradually as the release of the $OH^-$ by the fertilizer. The $Ca^{2+}$ of the lime promotes the agglomeration of soil particles through the ion bond bridge. $CaCO_3$ deposits would be formed through the reaction of part $Ca(OH)_2$ with $CO_2$ in the soil during alternation between drying and wetting. The $CaCO_3$ could act as a binder to join the soil particles together or as melt substances to coat or cover the aggregates, thus reducing the interaction between internal and external soil particles [48].

Comparatively, there was no $CaCO_3$ coating to be observed on the surface of the soil aggregates in the functional fertilizer treatments. The possible reason is that the functional fertilizers contain a lot of Mg in addition to Ca ions, which promote soil particle bonding and agglomeration. The Mg from the fertilizers will prevent $CaCO_3$ precipitation. In addition, the functional fertilizers had the ability to release calcium and magnesium slowly in the soil. The calcium and magnesium released gradually could fully bond with soil particles to construct water stable aggregates and good structure. Figure 4b showed that after lime application, obvious soil aggregates were produced, and the aggregates had obvious coating. However, obvious aggregates, more pore structures and the non- coating phenomenon were observed in the treatment of the functional fertilizers. The soil structure is also affected by Si. Porter reported that crystal size of goethite was decreased with the increase of the Si content, and the morphology changed from needle-like (Si free) to lamellar granular [49]. According to literature reports, when the addition of colloidal silicon

reached 8%, a bridge was formed between iron oxide and soil particles, at which time the stability of the aggregates was improved [50,51]. Therefore, Si in the functional fertilizers might play a similar role in the construction of the soil aggregates, but it needs further investigation.

## 5. Conclusions

In conclusion, the application of the functional soil amendment fertilizers is capable of construction of the stable aggregates and good soil structure. The elastic- and water-stable macroaggregates are increased and their dispersion rate is decreased. The mechanisms of construction of the stable aggregate are as follows. The functional fertilizers can gradually neutralize soil $H^+$ and prevent soil colloid dispersion of $H^+$ or mono-cations such as $K^+$ and $Na^+$. Soil particles are bound together to construct micro-agglomerates and then macro-agglomerates through the $Ca^{2+}$, $Mg^{2+}$ bond bridge and the $CaCO_3$, $MgCO_3$ salt bridge and the adhesion of SiO or $SiO_2$, $Fe_2O_3$, $Al_2O_3$ as well as the other amorphous substances from the functional fertilizers. The results can provide a theoretical basis for the sustainable development of agriculture by using functional fertilizers to improve soil, construct stable aggregates and form an ideal soil structure.

**Author Contributions:** Conceptualization and methodology, X.F., Y.Z. and J.J.; software, Y.Z. and J.F.; validation, Y.Z., J.J. and X.F.; formal analysis, K.J., and J.F.; investigation, Y.L., J.J. and K.J.; resources, X.F. and S.S.; data curation, Y.Z., J.J. and X.F.; writing—original draft preparation, Y.Z., J.J. and X.F.; writing—review and editing, X.F.; visualization, C.G., X.Q. and L.Z.; supervision, X.F.; project administration, X.F.; funding acquisition, X.F. All authors have read and agreed to the published version of the manuscript.

**Funding:** The authors are extremely grateful to China Agriculture Research System (CARS-31-07) and National Key Research and Development Program of China (2018YFD0201100) for providing research funding support.

**Institutional Review Board Statement:** Not applicable.

**Informed Consent Statement:** Not applicable.

**Data Availability Statement:** Data sharing is not applicable.

**Conflicts of Interest:** The authors declare that they have no conflict of interest.

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
