# Peer review of "Soil Aggregate Construction: Contribution from Functional Soil Amendment Fertilizer Derived from Dolomite"

_sustainability, doi:10.3390/su141912287_

Round 1
Reviewer 1 Report
The manuscript has well written but still needs revisions for consideration of publication in MDPI. Many of the suggestions have been rendered in the body of the text of the MS. It was not clear whether the study was conducted in the field or pot. Also, the design of the experiments was not properly explained. In some of the places, there was mention of organic fertilizers, organic may be manured, how come these be considered as fertilizers? Grammatical errors were also noticed and highlighted the manuscript in change mode. The reference style is not uniform, the authors are requested to strictly adhere to MDPI guidelines for the presentation of the references. The abbreviation is too many even in the abstract, these were used without their expanded meaning. This can be avoided for better readability and understanding of the findings. The conclusion part seems very hurridly written, please also modify the conclusion

Author Response
Q: Many of the suggestions have been rendered in the body of the text of the MS.
Repply: The specific corrections have done as followings according to reviewer’s suggestions which have been rendered in the body of the text of the MS.
Q: design of the experiments
Repply: the design of the experiments have been explained and rendered with red letters in the revised manuscript.
1. Line 61: “organic fertilizer”
Repply: In the discipline of plant nutrition organic fertilizer is manure. Manure is a kind of traditional fertilizer in the world. Of course, fertilizer includes organic and chemical (or inorganic) fertilizer. However, we accept reviewer’s suggestion, and use the manure instead of the organic fertilizer.
Q: Grammatical errors were also noticed and highlighted the manuscript in change mode.
Repply: The errors have been corrected as reviewer suggested.
Q: The reference style is not uniform
Repply: The reference format has been modified to the MDPI prescribed format.
Q:The abbreviation is too many even in the abstract
Repply: Abbreviations are explained as required when they first appear in the text.
Q: The conclusion part seems very hurridly written, please also modify the conclusion
Repply: The new conclusion is as followings.
In conclusion, the application of the functional soil amendment fertilizers is capable of construction of the stable aggregates and good soil structure. The elastic- and water-stable macroaggregate are increased and their dispersion rate is decreased. The mechanisms of construction of the stable aggregate are as followings. The functional fertilizers can gradually neutralize soil H+ and prevent soil colloid dispersion of by H+ or mono-cations such as K+ and Na+. Soil particles are bounded together to construct micro-agglomerates and then macro-agglomerates through Ca2+, Mg2+ bond bridge and CaCO3, MgCO3 salt bridge and adhesion of SiO, Fe2O3, Al2O3 as well as the other amorphous substances from the functional fertilizers. The results can provide a theoretical basis for the sustainable development of agriculture by using functional fertilizers to improve soil, construct stable aggregates and form ideal soil structure.
Reviewer 2 Report
The manuscript investigated the effects of soil amendments on soil aggregate construction. The results could provide some evidences for the filed practice management. However, the manuscript should be improved according to the current form. In the discussion, the difference of various functional fertilizers was not demonstrated clearly. The metal ions input with the functional fertilizers were not estimated prior to the experiment. Moreover, many flaws could be observed in the text.
L26, micaro-agglomerates?
L27, SiO?
L43 was ?
L46, the number of ?
L75, to improve soil acidification ?
L98, Nmin? should be full spelled in the first time.
L118, KNO3, superphosphate(?) and K2SO4
L155, equations (1) and (2) are wrong. Different from the literature [30].
L159, equation (3) is not clear.
Figure 1 and figure 2. IR and DR share the same data. Delete one of them.
Author Response
Repply:
Thanks for your suggestion. The authors have made corrections according to the reviewer’s suggestion in the revised manuscript in red letters. In the discussion section, the difference of the functional fertilizers has been discussed as followings.
The efficacy of the CDFF treatment was better than that of the UCDFF due to the difference of dolomite added in them. Jiang [27] reported that calcination could significantly increase the total alkalinity and short-term cumulative alkalinity of the dolomite. After the dolomite is calcined, the carbonate in the dolomite is transformed into CO2 and released into the air. The CaCO3 and MgCO3 in the ore are transformed into CaO and MgO . Relatively, the molar number of alkaline substances of the same mass of dolomite increases after calcination. MgO and CaO of calcined dolomite will react with soil water to produce Mg(OH)2 and Ca(OH)2, which contain a large amount of OH- and high short-term cumulative alkalinity. They are obviously capable of treating or improving soil acid. In addition, Mg(OH)2 and Ca(OH)2 can gradually release calcium and magnesium while neutralizing soil acidity. It provides calcium and magnesium nutrients for soil. Therefore, the CDFF is much more better than UCDFF.
The dolomite in test is natural pure dolomite which was collected from Jinfeng mine in Linwu, Chenzhou, Hunan Province, containing 33.12% CaO and 18.97% MgO. No other heavy metal was detected.
Q: L26, micaro-agglomerates?
Repply: It was misprinted. It has been corrected as macro-agglomerates.
Q: L27, SiO?
Repply: The main form of silicon in natural dolomite is SiO2. The SiO has been corrected as SiO2.
Q: L43 was ?
Repply: Has been corrected.
Q: L46, the number of ?
Repply: Has been corrected as quantity of.
L75, to improve soil acidification ?
Repply: Has been corrected as to neutralize soil acid
Q: L98, Nmin? should be full spelled in the first time.
Repply:Nmin is a well known abbrevation of mineral nitrogen, which is the sum of ammonium and nitrate. It has been corrected.
Q: L118, KNO3, superphosphate(?) and K2SO4
Repply: superphosphate has been corrected as Ca(H2PO4)2
Q:L155, equations (1) and (2) are wrong. Different from the literature [30].
Repply: The formula has been modified and references added in the revised manuscript.
Q: L159, equation (3) is not clear.
Repply: References have been added for description.
Q: Figure 1 and figure 2. IR and DR share the same data. Delete one of them.
Repply: The Figure 1 and figure 2 have been redrawn according to reviewer’s suggestion.
Reviewer 3 Report
In this article, the influence of functional amendment fertilizer on soil aggregate was evaluated and the microstructure were observed by the SEM and EDS. It is important and significant to conduct this work. It is a topic of interest to the researchers in the related areas. The research methodology is reasonable, but the paper needs some revisions before acceptance for publication. The following issues need to be addressed to improve transparency and communication of the results,
1. Line 14 the SEM, EDS are test method while dolomite-based functional soil amendment fertilizers are the parameters set in this research. It is not appropriate to put them together in one sentence. Also, what kind of dolomite-based functional soil amendment fertilizers should be presented in the Abstract. So, please rewritten this sentence.
2. Line 38 above five soil fertility factors? Which five factors? Also repeated at Line 45!!!
3. Actually, it is difficult to follow in Line 34-55, please rewritten.
4. what is functional soil amendment fertilizer? Please give a definition.
5. Line 87 The logic of the two sentences doesn't run smoothly.
6. what is the particle size distribution of UCDFF, CDFF, Lime and the soil used in this research? Have you considered the influence of the particle size of these material on the size distribution of soil aggregate? Also, what is the chemical composition of these material? Please clarify!!
7. The reason for conducting Exp 1 and Exp 2 is not sufficient, and the process is not clear.
8. The significance level should be added in the Fig 1 and 2. These information also should presented in the text.
9. Please check that the reference format is correct. Like ref. 15.
Author Response
Thanks for your suggestion. The authors have made corrections according to the reviewer’s suggestion in the revised manuscript in red letters.
Q1. Line 14 the SEM, EDS are test method while dolomite-based functional soil amendment fertilizers are the parameters set in this research. It is not appropriate to put them together in one sentence. Also, what kind of dolomite-based functional soil amendment fertilizers should be presented in the Abstract. So, please rewritten this sentence.
Repply: The sentence has been rewritten in the revised manuscript as followings. Scanning electron microscopy (SEM) and energy dispersive spectroscopy (EDS) were used to investigate effect of dolomite-based functional soil amendment fertilizers on soil structure.
Q2. Line 38 above five soil fertility factors? Which five factors? Also repeated at Line 45!!!
Repply: The five soil fertility factors are explained in Line 35 of the original manuscript. However, by response of the reviewer;s suggestion, we rewrite the line 34 and 35 as:
Soil is a medium of crop growth and it is also a place where five soil fertility factors such as water, nutrients, aeration, heat and microorganisms are exchanged.
Q3. Actually, it is difficult to follow in Line 34-55, please rewritten.
Repply: Thanks for the suggestion. The author has rewritten the paragraph in the revised manuscript in red letters.
Q4. what is functional soil amendment fertilizer? Please give a definition.
Repply: Fertilizers that have the function to neutralize soil acid in addition to supply nutrients. It has been explained in the manuscript.
Q5. Line 87 The logic of the two sentences doesn't run smoothly.
Repply: The Two sentences have been rewritten as followings according to reviewer’s suggestion.
The function of the functional soil amendment fertilizer should supply nutrients on one hand and neutralize soil acid on the other hand. The fertilizer should also possess the ability to help to construct good soil structure, especially to promote formation of the water stable aggregates. That is why dolomite based Ca-Mg-Si functional soil amendment fertilizers have been developed on the base of our previous study about the dolomite based soil conditioner [27,28].
Q6. What is the particle size distribution of UCDFF, CDFF, Lime and the soil used in this research? Have you considered the influence of the particle size of these material on the size distribution of soil aggregate? Also, what is the chemical composition of these material? Please clarify!!
Repply:They have revised and expressed clearly in the revised manuscript.
Q7. The reason for conducting Exp 1 and Exp 2 is not sufficient, and the process is not clear.
Ensure reproducibility of the test
Repply: We have explained the reason in the manuscript, which is to repeat the research and make sure the results are reliable.
Q8. The significance level should be added in the Fig 1 and 2. These information also should presented in the text.
Repply: The significance level has been added to the Fig1 and Fig 2.
Q9. Please check that the reference format is correct. Like ref. 15.
Repply: All of the reference format has been checked and modified to the MDPI prescribed format.
Round 2
Reviewer 2 Report
The manuscript was improved well, however, there are some flaws that should be corrected. Please check the manuscript thoroughly.
Note subscript/superscript.
Author Response
Dear Sir or Miss:
Thanks a lot. We have checked the manuscript thoroughly and corrected some the flaws in it according to your suggestion. The correction in the manuscript is in "Track Format". You will find them easily.
Reviewer 3 Report
The Authers revised well except some typographical problems.
Author Response
Dear reviewer:
Thanks a lot. We have checked the manuscript thoroughly and corrected some the flaws in it and typographical problems according to your suggestion. The correction in the manuscript is in "Track Format". You will find them easily.